# Biphenylene: A Two−Dimensional Graphene−Based Coating with Superior Anti−Corrosion Performance

**DOI:** 10.3390/ma15165675

**Published:** 2022-08-18

**Authors:** Ke Ke, Kun Meng, Ju Rong, Xiaohua Yu

**Affiliations:** Faculty of Materials Science and Engineering, Kunming University of Science and Technology, Kunming 650093, China

**Keywords:** biphenylene, first principles, anti-corrosion coatings, two-dimensional graphene-based materials

## Abstract

Metal corrosion can potentially cause catastrophic engineering accidents threatening personal safety; thus, coating protection is a tremendously valuable anti-corrosion initiative. Recently, biphenylene, a novel two-dimensional (2D) graphene-based material, has achieved a remarkable synthetic breakthrough; the anti-corrosion properties of biphenylene, with its specific pore structure, are predicted to be beneficial in applications of metal corrosion resistance. In this study, the anti-corrosion mechanism of biphenylene is deciphered utilizing first principles and molecular dynamics. The results suggest that biphenylene with tetragonal, hexagonal, and octagonal carbon rings supplies adequate sites for stable O atom adsorption. The charge transfer amounts of +0.477 and +0.420 e facilitate the formation of a compact oxygen-rich layer on the material surface to acquire outstanding anti-corrosion properties. The moderate wettability of biphenylene prevents the water-based solution from encroaching on the biphenylene coating and substrate. In addition, the intensive binding between biphenylene and the aluminum substrate strengthens the integration of the two heterogeneous structures with −413.7 and 415.5 eV, which guarantees the durable application of biphenylene coating.

## 1. Introduction

Metal materials have served a pivotal function in all stages of societal transformation, and diverse metal materials have emerged as an essential substance base for human advancement [1]. Annually, the economic and safety devastation from industry failure is tremendous due to the weakening of metal durability from corrosion [2,3,4]. The measures to prevent metal corrosion are primarily methods of chemical protection [5], material structure modification [6], and coating protection [7]. The chemical protection could eliminate the galvanic interaction that induces chemical corrosion [8], while the material structure modification improves the anti-corrosion ability by cooperating with low-activity metals [9]. However, both of these involve heavy maintenance efforts and severe substrate requirements, thereby failing to deliver integral protection. The coating protection completely isolates the metal from the corrosive medium with more generous selectivity of the substrate, which has resulted in extensive applications in anti-corrosion.

The coating protection emphasizes a sufficiently rigorous bonding of the coating to the metal substrate, preferably with the rejection ability to the corrosive medium [10]. In recent years, 2D carbon nanomaterials represented by graphene have proven to be an effective anti-corrosion coating due to graphene’s distinctive impermeability, chemical stability, and mechanical strength [11,12]. Prasai et al. [13] transferred multilayer graphene layers onto the target surface by annealing growth and mechanical transfer to construct a thicker and more robust coating, achieving efficient corrosion resistance against metal surfaces. Singh et al. [14] fabricated the graphene oxide–polymer composite coating on copper by electrophoretic deposition to gain more than three times higher corrosion resistance compared with the bare copper substrate. Wang et al. [15] predicted bottom–up-composable DHQ-graphene corrosion-resistant coatings with promising structural stability, metallicity, and O adsorption through simulations based on density functional theory (DFT) and ab initio molecular dynamics (AIMD). It is evident that graphene and its derivative 2D graphene-based materials have stable chemical attributes, solid mechanical strength, and remarkably significant corrosion resistance, making them ideal for anti-corrosion applications. However, their anti-corrosion properties are subject to adjustment due to the structure irrationality of some 2D graphene-based materials, enabling their low surface activity energy and accelerating cathodic corrosion [16,17,18]. Furthermore, the investigation of current 2D graphene-based materials is primarily at the predictive stage [15,19,20], with few structures materialized for synthesis [21]. Therefore, the exploration of reasonable 2D graphene-based structures and the consideration of experimentally fabricated materials with superior corrosion resistance is a competitively promising research direction.

Biphenylene with alternative tetragonal, hexagonal, and octagonal carbon rings was first predicted by Balaban in 1968 [22]. Fan et al. [23] achieved the synthesis of biphenylene, a non-benzene 2D carbon isomer with a large ring, with sp2-hybridized carbon atoms through an on-surface interpolymer dehydrofluorination (HF-zipping) reaction. Despite the accomplished experimental synthesis of biphenylene, its anti-corrosion properties have not been fully explored in practice. Biphenylene is hypothesized to have a special selective adsorption property differing from honeycomb six-membered rings based on its lattice structure [24], which provides a foundation to probe into its corrosion-resistant potential. Thus, there arose a necessity to decode the anti-corrosion mechanism of biphenylene coating to specify its application feasibility.

In the present work, analyses of the global energy minimization, structural stability, band distribution, and mechanical properties of pristine monolayer biphenylene were carried out from the first principles. Moreover, we measured adsorption sites, charge transfers, density of states, and diffusion energy barriers during O atom adsorption on monolayer biphenylene; this was accomplished to confirm whether the adsorption behavior was adequate to deliver compact oxygen-rich layers in anti-corrosion. Finally, the hydrophilicity and the mechanism of mechanical binding to the substrate were elucidated in detail. The results indicated that biphenylene is an outstanding anti-corrosion coating material with exceptional O atom adsorption, optimal mechanical bonding, and reasonable hydrophobicity; additionally, its particular anti-corrosion mechanism inhibits cathodic corrosion acceleration. This work offers a credible reference for diverse 2D graphene-based structural applications and pioneers a fresh orientation for the prospective anti-corrosion coatings design.

## 2. Computational Methodologies

All first-principles calculations based on DFT were implemented by employing Vienna ab initio simulation package (VASP) [25,26]. The projected augmented-wave approach (PAW) [27] and the exchange-correlation term of the generalized gradient approximation (GGA) [28] in the Perdew–Burke–Ernzerhof scheme (PBE) [29] were utilized in geometry optimization as well as in static self-consistent and non-self-consistent calculations. The cutoff energy of 500 eV and Monkhorst–Pack grids with 5 × 5 × 1 were installed as the main parameters [30]. Meanwhile, the convergence criteria for energy change and maximum force were set to 10^−6^ eV and −0.01 eV Å^−1^, respectively. Moreover, a 15 Å perpendicular vacuum layer was added for better sustainability of minimal interlayer interactions. In additional, the climbing image-nudged elastic band method (CI-NEB) [31,32] was used to compute the O diffusion energy barriers. Bader charge analyses were performed based on code developed by Henkelman et al. [33]. To further evaluate the effect of van der Waals (vdW) interactions on the strength of O atom adsorption on planar monolayer biphenylene, a DFT-D3 correction by Grimme [34] was adopted in correlative computations. Ultimately, the interface optimization was simulated with molecular dynamics simulations by utilizing the molecular dynamics package LAMMPS and verified with DFTB+ by employing Materials Studio.

## 3. Results and Discussion

### 3.1. Geometric Structure of Biphenylene

Biphenylene is composed of tetragonal, hexagonal, and octagonal pore carbon rings [23], as shown in Figure 1. The CIF files can be seen in Appendix A. There are two inequivalent carbon atoms (labeled C_1_ and C_2_) in this structure, occupying the 4z (0.199, 0.659, 0.500) and 2p (0.500, 0.160, 0.500) Wyckoff positions, respectively. We observed that biphenylene possesses a Pmmm symmetry (space group $D_{H1}^{2}$) by having an orthogonal lattice (a = 3.7630 Å, b = 4.5120 Å, c = 15.0000 Å) and six carbon atoms in a single cell. In addition, we measured different C–C bond lengths of d1 = 1.46 Å, d2 = 1.45 Å, d3 = 1.41 Å, and d4 = 1.45 Å in biphenylene with corresponding bond angles of θ_1_ = 89.99°, θ_2_ = 124.95°, and θ_3_ = 110.12°, which correspond to the previous data [35]. Obviously, the bond lengths of biphenylene are similar to graphene (1.42 Å), but the bond angles are drastically different from each other (e.g., the bond angle in graphene is 120°) [36]. The particular geometry exhibited by these bond lengths and bond angles credibly endows biphenylene with distinctive electronic properties and adsorption activity. Furthermore, we discovered that no band gap exists in the band structure of biphenylene and several bands cross the Fermi energy level, which is consistent with the published data [24]. This clearly illustrates the metallicity of biphenylene, and unlike graphene, no Dirac cone at the Fermi level (which acts as a type of crossing-point which electrons avoid) was observed; this agrees well with the experimental spectra previously reported [37]. To describe the electronic band structure more accurately, a hybrid functional (HES06) was used to reproduce it. It is evident that the hybrid functional results obtained here were consistent with those from the PBE band structure. As Mak et al. [38] found, the double bond in biphenylene, the 2s orbital, and two 2p orbitals are hybridized to reconstitute three equal sp2 hybrid orbitals. Additionally, each sp2−hybridized orbital contains 1/3s orbital component and 2/3p orbital component. Thus, the geometric structure of biphenylene would bring a diverse physicochemical modification environment, with more interesting properties to be expected. Moreover, as with many 2D carbon materials, the good structure and properties of the 3D stacked biphenylene warrant future investigation.

### 3.2. Mechanical Properties of Biphenylene

The mechanical response of materials within an elastic boundary is reflected by the elastic constants to adjudicate the mechanical stability. There are four independent elastic constants (*C*_11_, *C*_22_, *C*_12_, and *C*_66_) controlling 2D rectangular units. The relationship of strain energy and elastic constants for 2D sheets can be interpreted by the following equation [39]:(1)$$E_{strain}=\frac{1}{2}(C_{11}\varepsilon_{a}^{2}+C_{22}\varepsilon_{b}^{2})+C_{12}\varepsilon_{a}\varepsilon_{b}+2C_{66}\varepsilon_{ab}^{2}$$
where the strain energy in unit area is stated by *E_strain_*; the in−plane *a*− and *b*−axis strains, as well as the shear direction, are labeled as *ε*_*a*_, *ε*_*b*_, and *ε*_*ab*_, respectively. The elastic constants, *C*_11_ = 242 GPa·nm, *C*_22_ = 306 GPa·nm, *C*_12_ = 78 GPa·nm, and *C*_66_ = 81 GPa·nm, were derived by fitting the strain–energy curves. The biphenylene structure carries mechanical stability pursuant to the Born–Huang criterion for *C*_11_ × *C*_12_ − $C_{12}^{2}$ > 0, *C*_66_ > 0.

The mechanical properties of biphenylene are expressed in terms of in-plane Young’s modulus *C*(θ) and Poisson’s ratio *ν*(θ) [39]:(2)$$C(\theta )=\frac{C_{11}C_{22}-C_{12}^{2}}{C_{11}\alpha^{4}+\left[\left(C_{11}C_{22}-C_{12}^{2}\right)/C_{66}-2C_{12}\right]\alpha^{2}\beta^{2}+C_{22}\beta^{4}}$$
(3)$$\nu (\theta )=\frac{C_{12}\alpha^{4}-[C_{11}+C_{22}-\left(C_{11}C_{22}-C_{12}^{2}\right)/C_{66}]\alpha^{2}\beta^{2}+C_{12}\beta^{4}}{C_{11}\alpha^{4}+\left(C_{11}C_{22}-C_{12}^{2}\right)/C_{66}-2C_{12}\alpha^{2}\beta^{2}+C_{22}\beta^{4}}$$
where *α* = cos*θ*, *β* = sin*θ*, and *θ* is the angle along the a direction. 

As can be seen in Figure 2, the in-plane Young’s modulus along the a and b axes indicate the anisotropic mechanical properties of biphenylene. The calculated Young’s modulus and Poisson’s ratio for biphenylene are plotted in Figure 2. For 2D materials [40], it is clear that biphenylene presents obvious anisotropy in both the Young’s modulus and Poisson’s ratio, indicating good mechanical properties.

### 3.3. Single O Adsorption on Biphenylene

The ability to adsorb O atoms is a significant indicator in our research for oxygen-rich layer building. Therefore, the adsorption energy of one O atom on biphenylene surface was investigated. Considering the *Pmmm* symmetry of biphenylene, we were able to obtain nine potential adsorption sites in the 2 × 2 monolayer. In Figure 3a, the nine adsorption sites are attributed to three types of reaction positions, namely the top (T_1_ and T_2_), bridge (B_1_–B_4_), and hollow (H_1_–H_3_). The adsorption energy (*E_ads_*) of absorbed O atoms on biphenylene monolayer can be calculated from the formula [41]:(4)$$E_{\mathrm{ads}}=(E_{total}-E_{biphenylene}-n\cdot E_{O})/n$$
where *E_total_* denotes the total energy of biphenylene after adsorption of O atoms, *E_biphenylene_* is the energy of the original biphenylene without adsorption, *E_O_* is the cohesion energy of free O atoms, and *n* is the number of adsorbed O atoms. 

The adsorption energies of O atoms are shown in Figure 3b, and they range from −0.066 to −0.135 eV. The adsorption process is thermodynamically exothermic so that the adsorption energies display negative values. The lower the algebra values of the adsorption energy, the stronger the heat release, indicating that O atoms are energetically more favorable for stable adsorption. Among them, the adsorption energy of O atoms at the B_4_ site was optimal, which means O atoms preferred to adsorb here. It is generally recognized that the negative adsorption energy indicates spontaneous and stable adsorption of relative materials or adsorbing sites. As shown in Figure 3c, the O atom adsorption energies of biphenylene at all bridge sites were negative, and the adsorption behavior was superior to that of the majority of the structures compared. This stable and site-rich adsorption facilitated the formation of a compact oxygen-rich protective layer on the biphenylene surface to obstruct immediate contact between the metal substrate and corrosive media, which in turn provided bilateral protection against aggression.

### 3.4. Electronic Properties of Biphenylene

To probe the electron motion pattern of O atom adsorption on the biphenylene surface, we calculated the differential charge density of O atoms adsorbed on biphenylene to further understand the mechanism of adsorption. The differential charge density was calculated based on the following equation [41]: (5)$$\Delta \rho =\rho_{\mathrm{total}}-\rho_{O}-\rho_{biphenylene}$$
where *ρ_total_*, *ρ_O_*, and *ρ_biphenylene_* indicate the electronic charge distributions of O adsorbed on biphenylene, isolated O, and the pristine biphenylene, respectively.

Figure 4a,b show the differential charge density distribution consisting of O and C absorbed into bonds. Figure 4a depicts the charge density of O adsorbed at two C_1_, while Figure 4b shows that of O adsorbed at C_1_ and C_2_. The complete charge transfer can be seen in Table 1 and Table 2. The yellow and cyan zones denote the accumulation and depletion of charges, respectively. The charge distribution at the interface layer is presented in Figure 4c,d. We found the electronic details of the binding between the O atom and biphenylene, where the partial charge of O atoms transfers to the adjoining C atom, that is vulnerable to the charge distribution. Using Bader charge analysis, we determined that the number of electrons lost by O atoms upon adsorption at the B_1_ site is 0.477 e, while that of adsorption at the B_3_ site is 0.420 e. Thus, the charge transfer of the O atom adsorbed at the B_1_ site is 13.6% more than that adsorbed at the B_2_ site. Nevertheless, the average bond length of C–O in the adsorbed state in Figure 4b is 1.47 Å, which is shorter than that of 1.68 Å in Figure 4a; this indicates that there is not only physical adsorption between the O atom and the C atom on the biphenylene, but also likely chemical adsorption. Therefore, we can additionally confirm the superiority of biphenylene in adsorption performance.

We further investigated the density of states to fully evaluate the electronic properties of biphenylene. To accurately visualize the simulation results, the K−points were set to 10 × 10 × 1, amending the global setting of 5 × 5 × 1. The total density of states (TDOS) and projected densities of states (PDOS) for the systems in different adsorption states are depicted in Figure 5. The TDOS of biphenylene at the Fermi level was non−zero (Figure 5a,b), indicating biphenylene exhibited the electronic conductivity differently from two−dimensional semiconductor materials. Additionally, the p_z_−orbitals were dominant in all PDOS and the PDOS of C, and also non−zero at the Fermi level, thereby showing the metallic properties (Figure 5b,c). This conclusion again provided evidence for the origin of the intrinsic metallicity of biphenylene that we cited. Figure 5c,d illustrate the PDOS with adsorbed O atoms. The PDOS of C atoms remained non−zero and that of O atoms was zero at the Fermi energy level. It was evident that the electronic conductivity of biphenylene with adsorbed oxygen atoms contributed primarily via the C atoms, whereas the O atoms presented a conductivity inertness in interaction with biphenylene. Moreover, the density of states in adsorbing state were clearly higher compared with pristine state, reflecting a more aggressive free flow of charge around the C atom. In addition, biphenylene had a high DOS of 0.413 states per eV per atom at the Fermi level, which is similar to the previous data [24]. This value was not only five times larger than that of (4, 4) carbon nanotubes (0.7 states per eV per atom [42]), but also was larger than those of metallic net−W (0.12 states per eV per atom), DHQ−graphene (0.193 states per eV per atom [15]), and net−Y (0.15 states per eV per atom). It is reasonable to assume that the adsorbed O atoms combined with the C atoms to construct strong C−O bonds, which enabled the O atoms to accumulate stable oxygen-rich layers on the biphenylene surface. 

### 3.5. O Diffusion Behavior on Biphenylene

We further calculated the diffusion energy barriers of O atoms on various pathways utilizing the CI−NEB method to verify the anti−corrosion reliability of the biphenylene coating. It was noted that O atoms at the hollow sites on biphenylene migrated spontaneously to the top and bridge sites as complete structural relaxation proceeds. Thus, we analyzed the diffusion of O atoms only at the bridge and top sites of stable adsorption. There were four possible migration pathways of O atoms from one stable adsorption site to another on the biphenylene surface: green (Path 1), indigo (Path 2), blonde (Path 3), and peach (Path 4), as depicted in Figure 6a. The energy barriers of O atoms at different pathways characterized the migration efficiency here. Figure 6b–e detail the transition states of four paths with energy barriers. It was evident from the data that the migration difficulty of various pathways was ranked as Path 4 (1.548 eV) > Path 1 (0.405 eV) > Path 2 (0.076 eV) > Path 3 (0.036 eV). Of these, Path 4 offered the highest energy barrier to migrate, and the diffusion in this pathway was the most sluggish, which facilitated the bonding between O atoms and C atoms on biphenylene. The formation of C–O enhanced the combination of the oxygen-rich layer with the coating, thus ensuring the anti-corrosion ability of materials.

It was also noted that the diffusion energy barriers of O atoms on biphenylene were relatively low overall (compared with that of O atoms on graphene which is 0.74 eV [43]), and this favored the dispersion of O atoms to build a homogeneous coating. In particular, this higher energy barrier of Path 4 may be attributed to the robust adsorption of O atoms at the counterpart B_4_ adsorption site. In combination with the stable adsorption and the uniform distribution of O atoms, it is possible to enhance the densification of the oxygen-rich protective layer. Similar to metal oxide layers defending against further erosion, oxygen−rich layers can resist corrosion.

### 3.6. Wettability of Water Molecules on Monolayer Biphenylene

To investigate the corrosion protection of biphenylene in a humid environment, we simulated the wettability of water molecules on biphenylene by MD. We constructed a model for the simulation in which the upper layer was a spherical group of water molecules converging to simulate water droplets, while the lower layer was a biphenylene (or graphene) coating. The water droplets fell freely and came into mutual contact with the coating surface by gravity; thus, the wettability of biphenylene was analyzed by the density profile curve of water molecules. The relationships of water molecules density profile and interparticle distance of graphene and biphenylene are displayed in Figure 7a,b, respectively. As indicated, the horizontal distance between the water molecules and the center axis of the initial water droplet was used as the interparticle distance, and analysis steps with an interval of 1000 fs per step from 6000 to 25,000 fs were taken as the temporal analytic criteria. The density profile of biphenylene showed an analogous regulation to that of graphene. When the interparticle distance became larger, the density profile of water molecules gradually decreased. Except for a deviation from the initial contact curve, the water molecules density profile pattern smoothly converged to stability in the temporal analysis. The evolution of water droplet contact with the coating was visualized by exhibiting a side view of the dynamic process of the MD model. It is straightforward to view the incomplete spread of water molecules in contact with the biphenylene (or graphene) coating. In fact, the incomplete spread was effective in preventing the penetration of water-based corrosive media [44]. Therefore, considering the competence of graphene as an ideal coating material [45,46,47] and the consistency of wetting regularity between graphene and biphenylene, the wettability of the latter should be capable of accommodating the anti-corrosion goal as well.

### 3.7. The Binding Properties of Biphenylene with Aluminum Substrate

The binding firmness between the coating and the substrate is a crucial property in practical applications. To investigate the binding of biphenylene as an anti-corrosion coating to a metal substrate, we selected aluminum for our study, as it is susceptible to oxidation. The sectional views of graphene and biphenylene bind with aluminum (110) crystal face are exhibited in Figure 8a–c. It was clear that there were chemical bonds formed between the biphenylene and the aluminum substrate, whereas graphene maintained connection to the substrate only by interatomic van der Waals forces. This indicated that biphenylene provided a preferable integration to the aluminum substrate in anti-corrosion. We produced the binding models, where aluminum atoms bonded with octagonal (Bond 1) and tetragonal (Bond 2) carbon rings, respectively, as shown in Figure 8d,e. The transition layer dimensions for Bond 1 and Bond 2 were 1.81 Å and 1.82 Å, respectively, corresponding to maximum binding energies of −413.7 and −415.5 eV. It is generally recognized that high absolute values indicate a strong binding performance. As the data illustrated, the binding pattern of Figure 8c,e was marginally better than that of Figure 8b,d. Interestingly, Bond 1 and Bond 2 featured comparable transition layer dimensions and binding energy, which were superior to that of graphene (3.49 Å and −308.4 eV [48]) to reflect a stronger bonding of the biphenylene with the aluminum substrate. The robust bond of the coating to the metal substrate ensures the durability of the material and contributes to the sustainable anti-corrosion effect.

## 4. Conclusions

(1)The particular structure of periodic tetragonal, hexagonal, and octagonal carbon rings endows its stable mechanical properties, ample adsorption sites, strong O atom adsorption, and excellent electronic properties.(2)Biphenylene coating with a dense oxygen−rich layer and appropriate wettability could isolate the matrix from corrosive media.(3)The charge transfers of O atoms while adsorbing at biphenylene are +0.477 and +0.420 e, indicating the preferable adsorption properties in oxygen−rich layer construction.(4)The rigid binding of the biphenylene coating to the aluminum substrate with the energy of 413.7 and 415.5 eV enhances the durability of the anti-corrosion material and its resistance to external interference.(5)This paper systematically expounds the superior anti−corrosion properties of biphenylene and the potential to replace graphene as a better anti-corrosion coating. The research was based on theoretical calculation; our practical conclusions remain be verified in subsequent work.

## Figures and Tables

**Figure 1 materials-15-05675-f001:**
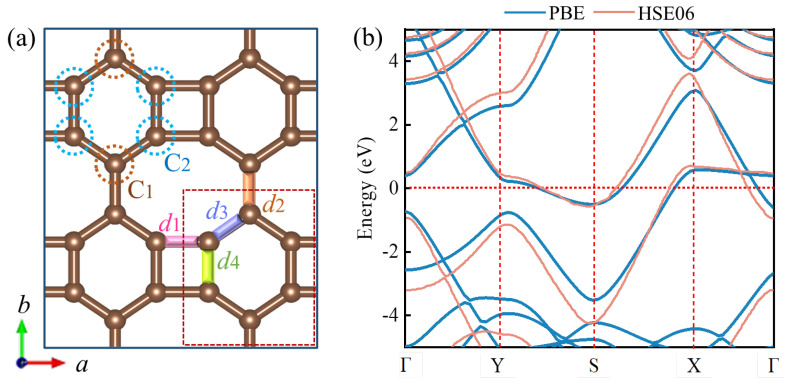
(**a**) Optimized structure of biphenylene. (**b**) The band structure of biphenylene calculated with PBE and HSE06.

**Figure 2 materials-15-05675-f002:**
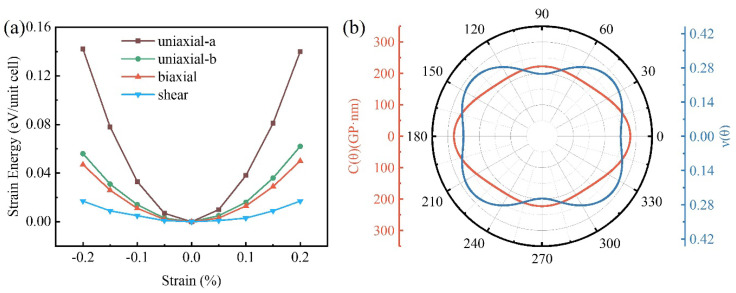
(**a**) Calculated deformation energies as functions of external strains for biphenylene. (**b**) The direction-dependent Young’s modulus *C*(θ) and Poisson’s ratio *v*(θ) of biphenylene.

**Figure 3 materials-15-05675-f003:**
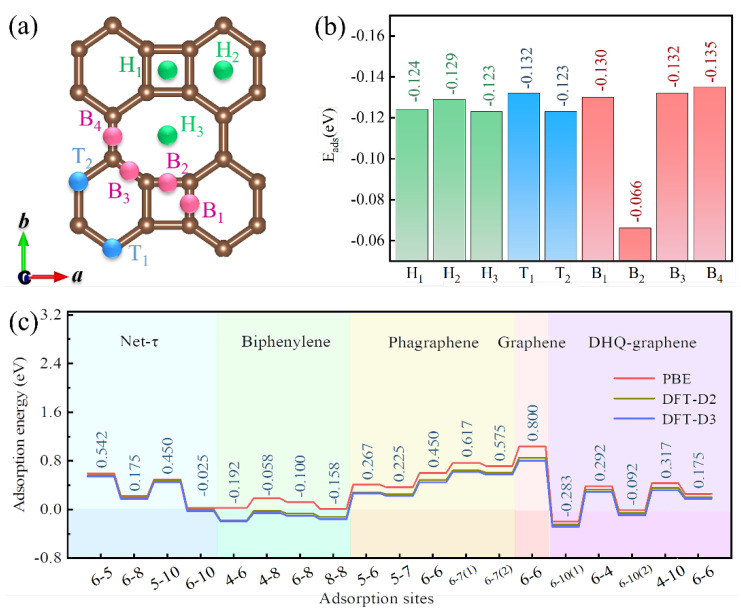
Adsorption properties of biphenylene. i−j, the bridge sites where i− and j−membered carbon rings intersect. (**a**) Adsorption sites of O atoms on the surface of biphenylene. (**b**) Adsorption energies of a single O atom adsorbed at different sites. (**c**) Adsorption energies of different graphene-based materials.

**Figure 4 materials-15-05675-f004:**
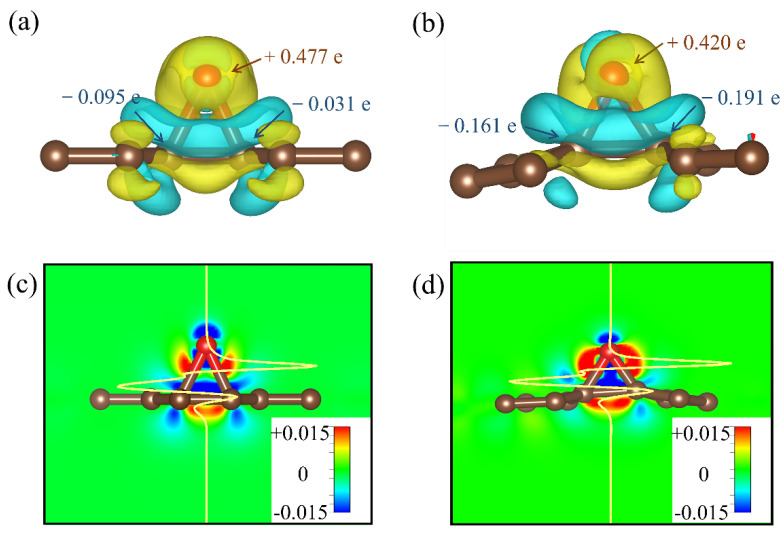
Side view of discrepancies in the charge density of O adsorption on different sites. Charge density of the lowest energy adsorption configuration for O adsorption: (**a**) two C_2_, (**b**) C_1_, and C_2_. Interfacial charge transfer for O adsorption: (**c**) two C_2_ and (**d**) C_1_ and C_2_.

**Figure 5 materials-15-05675-f005:**
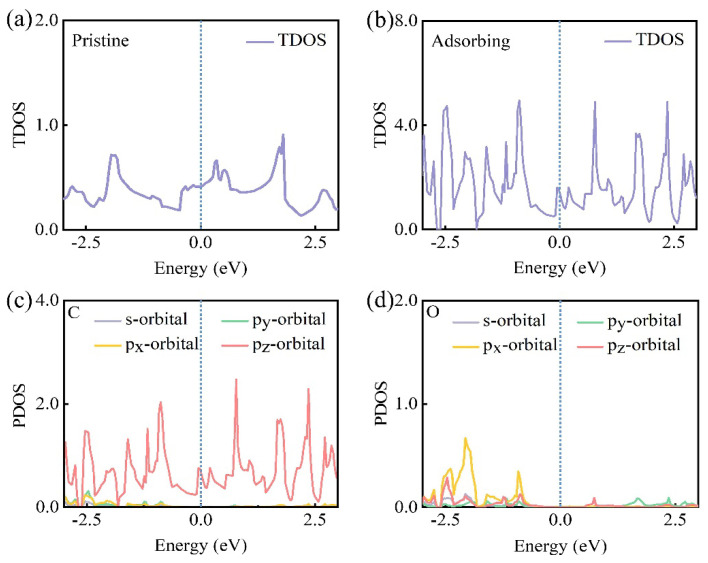
Electronic properties of (**a**) pristine structure and (**b**–**d**) adsorption structures. The TDOS of (**a**) pristine biphenylene and (**d**) adsorbing biphenylene. The PDOS of (**c**) C and (**d**) O.

**Figure 6 materials-15-05675-f006:**
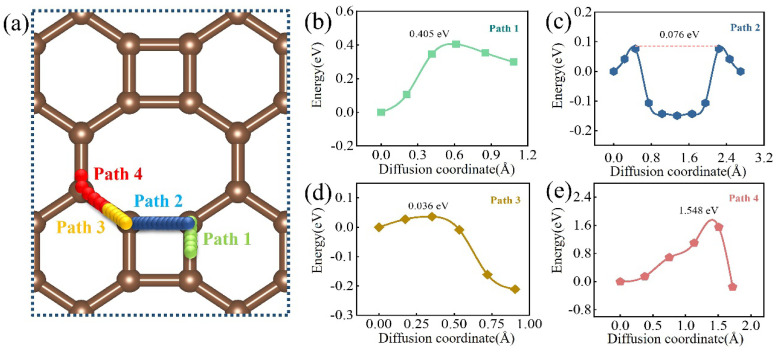
The diffusion behavior of O atoms on biphenylene monolayer. (**a**) O atom diffusion paths. (**b**–**e**) Minimum energy pathways and migration barrier of Paths 1−4.

**Figure 7 materials-15-05675-f007:**
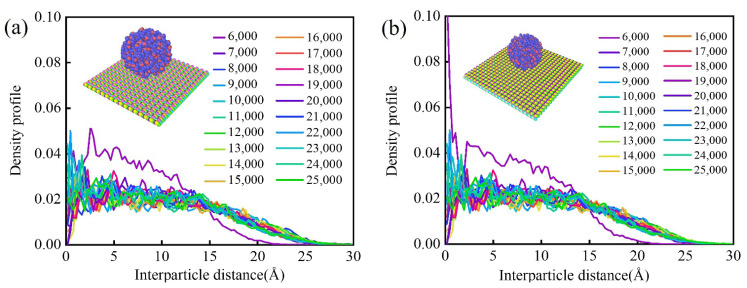
Wettability of water molecules on coatings. Density profiles of (**a**) graphene and (**b**) biphenylene.

**Figure 8 materials-15-05675-f008:**
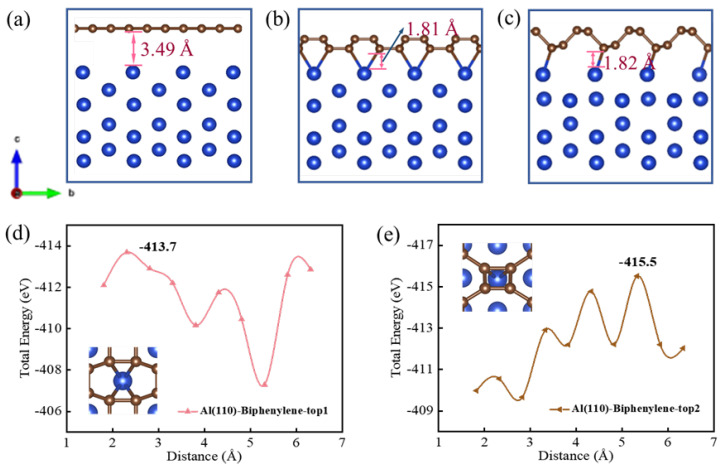
Binding properties of anti-corrosion coatings with aluminum substrate. (**a**) Sectional views of graphene bound with aluminum. (**b**,**c**) The Bond 1 and Bond 2 of biphenylene. (**d**,**e**) Energy variation of biphenylene in the coating−stripping process.

**Table 1 materials-15-05675-t001:** The charge transfer of the atoms of biphenylene while O atom adsorbing at B_1_ site.

C_i_/O	C_1_	C_2_	C_3_	C_4_	C_5_	C_6_	O
e	−0.141	−0.083	−0.031	−0.095	+0.147	+0.061	+0.477

**Table 2 materials-15-05675-t002:** The charge transfer of the atoms of biphenylene while O atom adsorbing at B_3_ site.

C_i_/O	C_1_	C_2_	C_3_	C_4_	C_5_	C_6_	O
e	+0.001	−0.191	−0.161	−0.001	−0.042	0.021	+0.420

## Data Availability

The data that support the findings of this study are available from the corresponding author, Meng Kun, upon reasonable request.

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
