# Peer review of "Biphenylene: A Two−Dimensional Graphene−Based Coating with Superior Anti−Corrosion Performance"

_materials, 2022, doi:10.3390/ma15165675_

Round 1

Reviewer 1 Report

The authors describes the application potential of biphenylene in anti-corrosion coatings, promising to expand the variety of 2D 21 graphene-based coatings to promote the significant reformation of metal anti-corrosion. The tackled research problem is interesting, however, it demands some additional comments.

·         The author's English writing should be further strengthened, and the expression of some sentences in the text needs to be further regulated.

·         The abstract must be concise and important findings (quantitive values) must be mentioned there.

·         The novelty of the paper should be better worked out.

·         Authors are advised to extend a little bit the presented research literature. There are some papers which falls into the scope of the article, which may be checked Please add new references to the manuscript.

·         Was the inhibition efficiency of Biphenylene checked by electrochemical measurements?

·         The Discussion section must be rewritten to be better understood by a broad audience and the conclusions in the manuscript should be more organized Please provide the implications of your results. A depth discussion on the anti-corrosion mechanism of Biphenylene would bring more value to your paper.

·         In general, the author should compare her results to those published in the literature and discussion them.

·         What are the most favorable adsorption sites of Biphenylene on steel or aluminium?

·         A 3D stacked structure of this material should be investigated.

·         All equations need references.

·         Please add strain energy and mechanical properties (Young’s modulus C (θ) and Poisson’s ratio) of Biphenylene.

Reviewer 2 Report

The manuscript presents an interesting study about biphenylene, a two-dimensional graphene-based coating, which can be used to improve the corrosion resistance of the substrate. The paper needs minor revisions before it is processed further, some comments follow:

The abstract must be improved. Please introduce the main conclusion of this study.

Figures 2, 4, 5, 6 are not clear, please replace it.

The conclusion section should be improved. Conclusions present some of the results discussed above in the paper with very limited discussion. Conclusions can be written with points. Also, write research suggestions and limitations.

Reviewer 3 Report

The authors of this study report the potential of biphenylene in anti corrosion coating
theoretically. This paper could be interesting to the readers of the journal, but not in its
present form. There are several unclear statements and grammatical errors scattered in
the whole ms.

My comments are as follows:

1. The abstract is not portraying the whole story. It must be rewritten with care.
Specifically, the last line of the abstract is totally unclear.

2. Biphenylene exhibits semiconductor to metal phase transition, and it is quite
natural to get the metallic nature using PBE functional. This is not at all a
surprising discovery of authors (line no: 117). There are several other reports
already published. I urge the authors to compare their own results with published
reports.

3. The authors should add the crystallographic data of the used structure to
increase the quality and validity of their results.

4. In line 121: "In this way, the geometric structure of biphenylene would bring
diverse physicochemical modification environment, thereby more exotic
properties to be expected." This part needs more clarifications. The band
structure must be discussed in in details. The roles of bonding environments, sp2
hybridization and appearance of dirac points etc. must be discussed.

5. In Figure 2c the authors compared the adsorption energies of several graphene-
based materials using several DFT functionals in terms of some numbers. What
is the cause of this comparison? The authors must explain the insight and
conclusion of this comparison.

6. The authors must rewrite the bader charge density analysis part. The figure 3
needs more clarification. The authors must provide a table with all charge density
values and provide a contour diagram to understand the charge transfer and
bonding environments. "...the Bader charge analysis, the number of electrons by
O atoms upon adsorption at ...." (line 170) is absolutely unclear.

7. How many K-points are used for DOS calculation? The authors must discuss the
origin of the peaks of DOS spectra and compare them with band structure.

8. There are several unclear sentences in section 3.5. The authors must rewrite this
part. They must explain the reason behind the difference between the spectra
obtained using graphene and biphenylene.

9. The conclusion part must be rewritten. It needs more clarity.

10. Overall, this ms has significant writing flaws. I urge the authors to check with a
native english writer before its resubmission to anywhere.

Round 2

Reviewer 1 Report

The authors have addressed the questions raised.

Best Regards

Author Response

Thank you very much for your contribution to the improvement of our manuscript.

Best regards!

Reviewer 3 Report

The authors of this study answered my comments and revised their ms to some extent. However, I think there are some minor points to be addressed before I recommend this paper for publication. My comments are as follows:

1.     Discussion about the band structure and DOS spectra and their origin are important for any phase change materials. If the authors have problems with length of the ms, they can opt for SI. I knew the material is not invented or reported for the first time by the authors. So, the authors must discuss and compare their results with previously published results. Did the authors perform any high level (e.g., HSE) calculationI think they must check their result with some high-level calculation. 

2.     Setting K-point to 0.01π is not a professional practice. I urge authors to use a dense K-mesh to reproduce the result and check the validity of their results. 

3.     The authors can provide directly a CIF file of their structure to increase the quality of the ms. Then the readers can easily reproduce the results presented by the authors.   
